# Experience, Principles and Parameters in the Sturgeon Quality Assessment by Anomalies in Early Ontogenesis (A Review)

**DOI:** 10.3390/biology11081240

**Published:** 2022-08-19

**Authors:** Saule Zh. Assylbekova, Ekaterina V. Mikodina, Kuanysh B. Isbekov, Gulmira M. Shalgimbayeva

**Affiliations:** 1Fisheries Research and Production Center, Almaty 050016, Kazakhstan; 2Russian Federal Research Institute of Fishery and Oceanography, Moscow 105187, Russia

**Keywords:** Acipenseridae, species, artificial reproduction, hatcheries, technology, Russia, Caspian states, sturgeon abnormalities

## Abstract

**Simple Summary:**

The artificial reproduction of endangered Sturgeon in the transboundary Caspian basin has been performed by five states (Russia, Iran, Kazakhstan, Azerbaijan, Turkmenistan). The Russian Federation is the Sturgeon breeding leader with the largest number of Sturgeon Hatcheries—eight, which produce and release up to 61 million Sturgeon juveniles into the natural habitat. This technology was systematized and published in the Caspian states national and international scientific press and FAO. However, this guide does not sufficiently reflect the presence of Sturgeon anomalies in early ontogenesis, identified in the second half of the 20th century. The change in Sturgeon breeding paradigm from wild producers to brood stock spawners provides the incentive and basis for detailed study of Sturgeon anomalies. All previously known Sturgeon anomalies are divided into nine classes. Assessment of Sturgeon anomalies as predictors of their survival in the natural habitat will allow the evaluation of the Sturgeon stock status in the Caspian Sea.

**Abstract:**

Purpose: Review the experience, principles and parameters of the sturgeon assessment quality by anomalies in early ontogeny. Results: Maintaining the number of sturgeon fish in the transboundary Caspian Sea is provided by five states (Russia, Iran, Azerbaijan, Turkmenistan, Kazakhstan) at 16 Sturgeon hatcheries, where their artificial reproduction is carried out. FAO recognizes Russia’s leadership in creating the basic technology for the sturgeon artificial reproduction, but the other four Caspian states also make a significant contribution to its modern optimizations. There is almost a century of tradition behind the technological development of artificial reproduction in sturgeons. During the artificial reproduction of sturgeons, anomalies in the structure and functions may occur, such as deformities, defects in organs and tissues, edema, hematomas, etc. The sturgeon anomalies classification is based on structural and functional principles. Identification of anomalies is carried out on the basis of a previously created classification, divided into nine large classes. Identification of sturgeon anomalies during the period of their artificial reproduction makes it possible to clarify the real value of replenishment of their stock. Methods: Analysis of professional scientific literature and practical guides. Conclusions: The presence and number of Sturgeon anomalies make it possible to determine their death percentage to calculate the real number of replenishment of the Caspian Sturgeon stock.

## 1. Introduction

Sturgeon (Acipenseridae) appeared about 250 million years ago after large-scale tectonic, geological, and climatic changes on the face of the Earth and evolutionary transformations of its biota [1]. The first fossil remains of the ancient sturgeon ancestors were found at the beginning of the Triassic period of the Mesozoic Era which lasted 250 million years. Their evolution was found during 56 million years of the Jurassic and the subsequent 79 million years until the end of the Cretaceous period of Earth’s geological history [2,3,4,5]. Over the next 186 million years, the sturgeons reached their zenith and since then till present time they have retained a number of primitive structural features—cartilaginous skeleton, lifelong preserved chord, presence of only upper and lower arches instead of vertebrae, heterocercal tail, spiraculum, arterial cone in the heart, spiral valve in the intestine, rudiments of ganoid scales in the form of a bony “plate” and others [2]. The oldest known sturgeon is *Chondrosteus acipenseroides* from the Early Jurassic Era [6].

Beginning in the second half of the 20th century, the number of sturgeons and their range began to decline rapidly and now these fish are “rare, endangered, and disappearing”. However, in a gastronomic sense, they remain a “delicacy” and “gourmet” product and are in a high demand among consumers.

It is reasonably assumed that several species of Sturgeon, for example, the Chinese paddlefish or *Psephurus gladius* from the Yangtze River (People’s Republic of China), Syr Darya Sturgeon *Pseudoscaphirhynchus fedtschenkoi* (Kessler, 1872) from the basin of Syr Darya River (Kazakhstan, Uzbekistan), as well as the Beluga *Huso huso* (Linnaeus, 1758) (Grate or Giant Sturgeon) from the Azov, Black Sea and Caspian populations, have not been found in their areal domain for several decades [7,8,9,10,11]. It is recognized that since 2009, the Beluga has practically no longer been reproduced in the wild not only in Russia, but also in Kazakhstan due to a decrease in the number of natural spawning grounds and a low efficiency of reproduction in nature. For example, since 2010, only single specimens of Beluga have entered the Ural River annually [12].

The main way to preserve the population of Beluga and other sturgeon species in the wild is through artificial breeding in hatcheries with a subsequent release of juveniles to empty niches of the natural habitat [9,13,14,15,16,17]. Recognition of the fact that Sturgeon fish are close to extinction, served as the basis for inclusion of the majority of Sturgeon species or its populations in the international ICUN Red List [18].

None of the global natural disasters (transgressions and regressions of ancient marine areas, glaciation, etc.) could lead to the current position of Sturgeon as species close to extinction as overfishing [1,11,17].

An effective way to maintain the number of Sturgeons in the natural habitat is through artificial breeding. The goal of this work is to characterize sturgeon hatcheries in the countries of the Caspian region, the condition in which the sturgeon is artificially bred, and types of anomalies in their structure and functioning. 

## 2. Results

*Some information on sturgeon hatcheries of the Caspian countries.* Already from the beginning of the first third of the 20th century, biological scientists and the societies of countries where there were numerous sturgeons, especially in the former USSR and modern Russia, determined the problem of a rapid decline in sturgeon numbers which initiated long-term studies of their reproductive biology and the development of approaches to reproduction under human control [19,20,21,22,23,24,25,26,27].

In countries with access to the Caspian Sea (Russia, Iran, Turkmenistan, Azerbaijan, Kazakhstan, after the start of construction of hydroelectric dams) on the rivers flowing into it—spawning beds of breeders and emigration of juvenile sturgeon, the hatcheries were organized, and biological bases and fish breeding technologies were developed to maintain the number of these fish [28,29,30].

The situation was mentioned for the first time by the Russian Federation (former USSR) in the middle of the 20th century. In the 1960s, the design, construction and commissioning of eight federal sturgeon hatcheries (SH) commenced on the Lower Volga that engaged in the artificial reproduction of sturgeons (Table 1). At present, we can mention six hatcheries of the aforementioned hatcheries [31]. Subsequently, private hatcheries were built. Simultaneously with federal Sturgeon hatcheries on the Lower Volga, more were built in other regions of Russia: Temryuksky SH (1964, Krasnodar Territory), Grivensky SH (1972, Krasnodar Territory), Konakovsky SH (1973, Tver Region), Donskoy SH (1995, Rostov Region), Selenginsky Experimental Sturgeon Hatcheries on the Lake Baikal (1986, Pribaikalsky District), Vladimirovsky SH (2008, Smidovichsky District of the Jewish Autonomous Region) and many others. 

At present, in the Caspian Sea basin, sturgeon hatcheries are located not only on the Lower Volga, but also in other regions, including in the zones of national jurisdiction of the other four Caspian countries mentioned above. In the Republic of Kazakhstan, sturgeon hatcheries appeared later than in other countries of the Caspian region. The first of them—Ural-Atyrau SH which was built in 1985, the second—Atyrau SH built in 1998. The source of sturgeon breeders for sturgeon artificial reproduction in Kazakhstan is the pre-estuarine space and lower reaches of the Zhaiyk River (Ural—*rus*). However, since 1990, a decrease in sturgeon stocks has been noted which are represented mainly by sturgeon and single specimens of Russian Beluga Sturgeon and Fringebarbel *Acipenser nudiventris* (spike—*rus*) [19,32].

*History and experience in the development of S**turgeon artificial reproduction technology.* Priority in creating the Sturgeon artificial reproduction technology including all its most important stages—stimulation of maturation, gametes production, fertilization, polyspermy, egg incubation, cultivation of prolarvae, larvae and fry—undoubtedly belongs to Russia [40]. This is due to the fact that by the beginning of the 20th century European countries had practically lost the possibility of industrial sturgeon fishing. In Russia, the number of sturgeon populations was sufficient until the beginning of the 21st century, and commercial fishing continued until 2010.

Russian fundamental scientific studies of early Sturgeon ontogeny biological features were started as early as 1930–1940. Later, their results were the emergence of two new Russian activities in Sturgeon conservation. The first direction was a fundamental scientific Sturgeon biology study, the second—their practical application. The fundamental scientific direction was named the «School of Developmental Biology». Its founders in Russia were Russian (Soviet) researchers such as T.A. Detlaff, A.S. Ginzburg, G.M. Ignatieva, N.L. Gerbilsky, and O.I. Schmalhausen [19,20,21,22,23,24,25]. These scientists studied the structure and functions of different species of sturgeons in early ontogenesis (Russian Sturgeon, Large or Giant Sturgeon (Beluga), Persian Sturgeon, Spike Sturgeon, Sevruga and some others) with the publication of original scientific drawings, photographs and temporal patterns of sturgeon development, namely biological age. To determine the biological age, a relative dimensionless unit for assessing developing sturgeons was created and called “tau-zero” (τ_0_), later—“detlaff” [22,25]. The relative dimensionless unit of the biological age of poikilothermic objects — “tau-zero”, was first used to control and compare the timing of development of different sturgeon species; a relative dimensionless characteristic was developed to determine the duration of development at different stages. Later, other relative dimensionless indicators of development were proposed by the world scientific community. However, the foundation was laid in Russia.

Based on the Sturgeon study predominantly, the results in former USSR were published in several academic monographs on the early ontogenesis of fish [19,20,21,22,23,24,25]. Some of them have been translated into English and became the main guide for the development of Sturgeon artificial reproduction in other countries [20,25]. Let us remind the reader that the formation of Sturgeon artificial reproduction technology in Iran took place with the support of Russian scientists and practitioners too. The books listed above have acquired the status of “desktop” not only among scientists involved in the biological reproductive study of Sturgeon fish, but also among practitioners from Sturgeon hatcheries. Many students and followers of these scientists continue to work in this field. 

It should be borne in mind once again that the described scientific results were obtained in a former state called the Soviet Union. It included such now independent states as Ukraine, Kazakhstan and Azerbaijan, where Sturgeon artificial reproduction technology was carried out. If the scientific and technological foundations of Sturgeon artificial reproduction were created by scientists from Russian scientific institutes belonging to the Russian academy of sciences, then the practical verification of the data obtained was carried out directly in the conditions of the Kurinsky Sturgeon hatchery in Azerbaijan when it was a republic of the former USSR. Now, Azerbaijan is an independent state—the Republic of Azerbaijan. 

In the USSR, the state, communication, and scientific language was predominantly Russian. That is why the basic scientific and practice of sturgeon aquaculture literature and guides were mostly originally in Russian. 

The need for fundamental and practical information about the early development of sturgeon is becoming especially relevant at the present time, when five Caspian states have established a ban on sturgeon fishing in the Caspian basin. Global demand for Russian Sturgeon technologies is clearly demonstrated by the Guide to Artificial Reproduction of Sturgeon issued by the Food and Agriculture Organization as Fisheries Technical Paper (Guide No. 558), not only in Russian but also in English. Sturgeon federal hatcheries and private fish farms of a number of Caspian countries excluding receipt and release of juveniles into the Caspian Sea are additionally engaged in commercial cultivation [25,27,29,41]. 

Due to the decline in the number of Caspian sturgeons and the ban on catching spawners in the Caspian, the formation of brood stocks has become a technological innovation. Practically, this happens in two ways: domestication of wild producers and growing them “from eggs to eggs”.

Breeding of brood stock producers from fertilized eggs purchased from other producers (countries) is common in the Caspian sturgeon states, but is fraught with some complications if purchased from irresponsible (unscrupulous) farmers. Herewith, the number of domesticated individuals as a source of commercials is extremely limited which can lead to a decrease in genetic diversity of the brood stocks. In addition, imported fertilized eggs in some cases are of hybrid origin which excludes the possibility of using the breed of such individuals for stocking and release. In this regard, it is necessary to apply modern methods of genetic study and monitoring of natural Sturgeon populations and their artificial brood stocks [26,27,42,43,44]. For Kazakhstan, as a Caspian state, the solution of this issue is extremely relevant [28,44]. Genetic certification of spawners and genotyping of artificial juveniles are carried out/introduced at Sturgeon hatcheries in these countries [29,45].

In early ontogenesis of the fish due to formation of fundamentally new morphological and functional properties (for example, changes in hormonal status, structure, behavior, type of nutrition), the juveniles transition from each embryonic–larval developmental stage to the next. For sturgeon, like for other fish species, this is a complex multifactorial process called the «critical period» [46,47,48]. In nature, the critical period of fish in the postembryonic period occurs in relatively adequate conditions of spawning and post-spawning niches. However, this does not exclude the ecological parameters and fluctuations of the habitat, as well as the food disadvantage and the impact of enemies. The impact of these factors leads to the death of part of the offspring, and the appearance of anomalies, which reduces the efficiency of natural reproduction. Despite the fact that the cultivation environment factors at Sturgeon hatcheries maximally imitate the natural factors, they are not completely identical to them, which allows us to state that the life of the reared juveniles and their transition through critical periods occurs under conditions of ecological deprivation. Mostly, this circumstance leads to the emergence of different structural anomalies in some parts of released juveniles and cultivating sturgeon brood stocks [49,50] resulting in a decrease in their adaptive capabilities and, as a consequence, the quality of embryos, prolarvae, larvae, fry and juveniles. This is exacerbated by the known diversity of fish in early ontogenesis [9,51,52,53,54], the study of which allowed the establishment of the criteria for estimation of their quality [55]. Wherein, for the last almost 40 years in Russia, there have been regulatory requirements for abiotic conditions of artificial reproduction and a standard for the weight and size of sturgeon juveniles released by hatcheries into basic rivers [56]. However, there are no requirements for phenotypic indicators.

*Principles of classification of Sturgeons anomalies.* At breeding and cultivation in Russian hatcheries and, as it was stated above, in deprived conditions, sturgeons have different morphological and functional anomalies: structure of olfactory organs, organs of vision, oral cavity, underdevelopment of fins or presence of an additional pair of fins, shortening of the gill cover, lack of tail, scute, curvature of the body (scoliosis, lordosis), as well as dropsy, tympania, decrease in the activity of a number of enzymes such as oxyreductase (dehydrogenase, alcohol dehydrogenase, lactate dehydrogenase, malate dehydrogenase, etc.), hatching enzyme, circulatory system, digestive enzymes, and other functional indicators [9,20,23,40,49,57,58,59,60,61]. The classification of Sturgeon anomalies, both in terms of the artificial breeding and fish of natural habitat, is based on the structure of body and organs, and their functional role in the organ system. The classification was provided in full for the first time in 2004, and further supplemented by other publications on this issue not only by Russian scientists, but by global scientists, for example, from the People’s Republic of China. Practicing fish farmers constantly accumulate new information during the breeding process, the majority of such information is their private know-how. 

*Morphological abnormalities* in the sturgeon body structure are the most noticeable and occur more often than in other cases (Table 2, Figure 1 and Figure 2). The percentage of such abnormalities varies depending on the fish age, period of detention in artificial conditions, type of fish farming, on average ranging from 5.5% but sometimes reaching 100% in the larvae [9,49,50,51,52,62,63,64]. At the same time, a deviation rate of up to 6% from the normal structure is considered normal [63]. At the sturgeon hatcheries of the Republic of Kazakhstan, accounting and analysis of anomalies in the structure of the produced juveniles has not been carried out to date.

It is widely known that abnormalities in fish structure often occur in the early ontogenesis during so called “critical periods”. There is a large amount of information about critical periods in embryos and sturgeon larvae [46,47,48,64]. A detailed analysis of the causes of critical periods in fish leading to abnormalities in the structure and functioning was published by N.G. Zhuravlyova [65] who divides them into two groups: non-optimal abiotic (temperature, oxygen saturation, salinity, illumination, depth of fish tanks, watercourse speed) and biotic conditions (malnutrition). The consequences of the first are a disbalance of the dynamics of neurohumoral regulation of form-building processes, autoimmune reactions of morphogenetic processes in the embryonic–larval and metamorphose periods, dysfunction of the hatching glands, the second—overflow of food in the intestines, gases of swim bladder in bony fish leading to deformation of the notochord with subsequent transition to lordosis or kyphosis of the spine, starvation, destructive processes or degeneration of organ and cellular structures of the digestive tract, dropsy of fin border, heart bag, ascites of the abdominal cavity, and traumatic objects of nutrition [66].

Based on the division of critical periods due to reaction place or impact on cells, tissues, organs, and the body, generally nine critical periods for fish are distinguished: 1. gametogenesis [66]; 2. fertilization; 3. gastrulation; 4. notogenesis; 5. histogenesis [67,68]; 6. organogenesis, 7. embryo hatching, 8. transition to mixed, then exogenous nutrition; 9. metamorphosis (where applicable) [65].

Knowing periods of hypersensitivity (critical periods) is necessary to create optimal conditions during the incubation period of eggs, i.e., embryo development, aging of prolarvae and larvae, rearing juveniles in order to increase their survival since the main deterrent factor to the sustainable development of aquaculture is the impossibility of obtaining 100% of viable juveniles on an industrial scale.

Due to the absence of adequate conditions in aquaculture, including parts of the artificial reproduction process, and of most of the factors of natural selection, the influence of which is experienced by natural populations (competition in nutrition, predator press, environmental conditions), some of the fish involved in artificial reproduction have various morphological anomalies.

The anomaly types are summarized in “Atlas of Abnormalities in gametogenesis and structure of sturgeon”, later supplemented and republished in English in Germany [9,49].

*Sturgeon anomalies classification.* The authors mentioned above analyzed the research results of many followers, and divided abnormalities in sturgeons into nine large classes with their own subsections:
Abnormalities in the structure of gametes and gonads.
1.1.types of abnormalities in the structure of mature eggs of sturgeon [19,50]1.2.disturbances in ultrastructure of the membranes of developing sturgeon eggs [21]1.3.types of abnormalities in the structure of the Siberian sturgeon eggs in different periods of gameto- and gonadogenesis types of abnormalit {[[ies in the structure of male gonads [68].
Types of abnormalities in the structure of larvae.
2.1.body shape abnormalities ([49,50], Figure 1 and Figure 2)2.2.abnormalities in the structure of external organs [9,49,65]2.3.abnormalities in the structure of internal organs [66]2.4.cell structure abnormalities [49]2.5.functional abnormalities in larvae development (hematomas, timpania, dropsy, lipid metabolism disorders [40])2.6.mechanical damage [49]
Types of abnormalities in the juveniles’ structure.
3.1.abnormalities in the structure of external organs: shortening of operculum [69]3.2.abnormalities in the structure of olfactory organ: non-union of the septum of olfactory organ, absence of olfactory epithelium, abnormalities in the structure of olfactory rosettes [69]3.3.abnormalities in eyes structure: pupil size decrease, local or diffuse cataracts, clouding of the cornea, abnormalities in the lens fractal structure, abnormalities in the structure of the inner parts of the eye.
Physiological disorders [70,71,72,73,74,75].Pathologies of muscle tissue [67].Functional disorders [59].Biochemical disorders [75].Hormonal disorders [76,77].Behavioral disorders [78].


It is believed that teratogenesis is most often found in the sturgeons cultivated precisely in industrial hatcheries (abnormalities of olfactory organs, organs of vision, oral cavity, underdevelopment of pectoral fins, shortening of the gill cap, lack of tail, scute, scoliosis, dropsy, presence of an additional pair of fins) which does not exclude the possibility of their subsequent influence not only on the survival and quality of the brood stock but also on their genetic constitution [43,45,57].

The appearance of abnormalities indicates the need to adjust artificial reproduction and breeding work especially the control over the selection of breeder pairs both caught in nature and from brood stocks. Genetic certification of brood stocks enables fish farmers to compile optimal pairs of breeders from available individuals in order to preserve a high genetic diversity of the natural population and increase viability of the offspring. It is necessary to continue monitoring the morphological and functional state of sturgeon in early ontogenesis as a measure of assessing quality of the used breeders and predicting the offspring survival.

A total paradigm changeover of the activities of Volga Sturgeon hatcheries to the work with spawners of the brood stock rather than wild breeders in recent years has begun at sturgeon hatcheries where the breeders are domesticated and brood stocks are formed from them or by “from eggs to eggs” method. Utilization of fish with structural abnormalities during formation of the brood stocks and work with them was also previously recommended due to their shortage, and such fish began to be included in already formed brood stock when the stock is replenished not only in Russia but also, for example, in Poland [79] or Spain [80].

## 3. Final Remarks

The Russian Federation, as one of the five states-users of the transboundary Caspian Sea and its Sturgeon stocks, has the largest number of Sturgeon hatcheries, where it produces and releases into the natural habitat the largest number of Sturgeon juveniles—about 61 million fish. This number is an order of magnitude higher than the number of juveniles released by other Caspian countries. The artificial reproduction of sturgeon juveniles is carried out on the basis of Russian national guidelines that are used by other nations. The available guidelines do not take into account the abnormalities of artificial sturgeon fry, which affect their survival and, ultimately, the abundance of stock in the Caspian Sea. The presence and types of anomalies in the structure and functions of sturgeons have been studied in detail and published mainly by Russian researchers [33,34,49,50,51,56,57,60,62,70]. The studies of Sturgeon abnormalities at hatcheries in the Caspian Sea basin countries were conducted mainly in Russia while in other countries of the region—Iran [36,37], Azerbaijan [35,36,37] they were minimized; in Turkmenistan are absent. We have found some data published on the presence of the types and number of anomalies in sturgeons at Kazakhstan Sturgeon hatcheries in the available literature [33,44]. The availability of this information is especially relevant for the Republic of Kazakhstan, since this country has the least experience in the artificial reproduction of sturgeons due to the relatively short operating time of national Sturgeon hatcheries. The Sturgeon artificial reproduction in Kazakhstan is based on the available Russian technologies and guidelines, but anomalies are not accounted for. A national guideline for the artificial reproduction of sturgeon has not yet been developed. It seems appropriate to start such studies with the subsequent publication of the manual. It can be national “Methodological recommendations for sturgeon abnormalities accounting in Sturgeon artificial reproduction hatcheries in the Republic of Kazakhstan to predict the survival of offspring”, the development of which shall be required. Regular field monitoring of the quality of sturgeon offspring in the process of artificial reproduction is necessary in order to predict survival and to maintain their numbers in the Caspian Sea.

## Figures and Tables

**Figure 1 biology-11-01240-f001:**
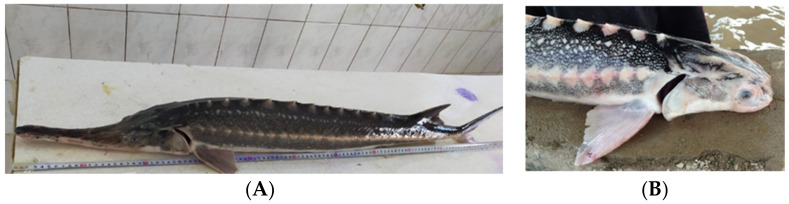
Stellate sturgeon *Acipenser stellatus*, Ural-Atyrau sturgeon hatchery (Kazakhstan), 1+ age, (**A**) standard fingerling morphology, (**B**) absence of rostrum due to mechanical damage.

**Figure 2 biology-11-01240-f002:**
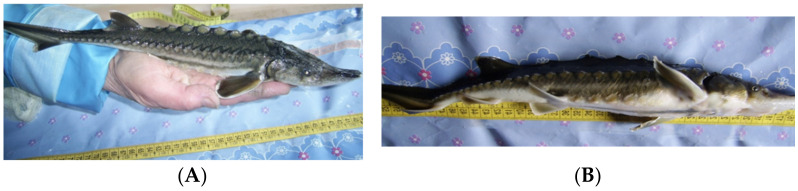
Sakhalin sturgeon *Acipenser mikadoi*, Okhotsky sturgeon part of salmon hatchery (Sakhalin island), 1+ age, (**A**) standard fingerling morphology, (**B**) tail downward curvatures and anal fin mechanical damage.

**Table 1 biology-11-01240-t001:** Sturgeon hatcheries of the Caspian states (Russia, Azerbaijan, Iran, Turkmenistan, Kazakhstan).

Country	Name of Sturgeon Hatchery (SH) *	Year of Construction/Base River/Location	Breeding/ Cultivation Object	Volume of Production/ Release, mln pcs	Reference
Russia	1. Kizansky	1955, Volga R., Kizan village	*Acipenser gueldenstaedtii,* *A. stellatus*, *Huso huso*	61	[31,32]
	2. Volgogradsky	1960, Volga R., Volzhsky City			
	3. Bertyulsky	1961, Volga R., Algaza village			
	4. Sergiyevsky	1963, Volga R., Sergiyevka village			
	5. Ikryaninsky **	1962, Volga R., Ikryanoye village			
	6. Lebyazhy	1979, Volga R., Narimanov City			
	7. Alexandrovsky	1974, Volga R., Trudfront village			
	8. Zhitninsky	1981, Zhitnoye village			
Azerbaijan	1. S. Kerimov (former Kurinsky)	1954, Kura R., Yenikend, Neftechala region	*A. stellatus,*	15 mln pcs	[33]
	2. Khili	2003 Neftechala region, Baku	*Huso huso*		[34]
	3. Shirvan	Shirvan City	Acipenseridae		[35]
Iran	1. Sefidrud	1971, Sefid-Rud R.	*Acipenser gueldenstaedtii*, *A. persicus*, *A. stellatus*, *A. nudiventris* *A. stellatus Huso huso*	3.5 mln pcs	[36,37,38]
	2. Shahid Marjan	1987, Gorganrud R., 45 km northeast of Gorgan		4–5 mln pcs	
Turkmenistan	Hazar Balyk	2015, coast of the Caspian Sea, near Turkmenbashi City	No data	100 tons of sturgeons, 2 tons of black caviar, 10 mln. cans/year	https://hazarbalyk.com/index.php (accessed on 12 February 2022)
Kazakhstan	1. Atyrau SH	1998, Yaitsky branch of the main channel of Ural River, between Rakusha and Yerkinkala villages, 10 km from Atyrau City (previously Guryev), 7 km from the coast of the Caspian Sea.	*Acipenser gueldenstaedtii* *A. stellatus*, *Huso huso*	7 mln pcs	[19,32,39]
	2. Ural-Atyrau SH	1985, Zolotoy Branch of Ural River, Zarosloye village, 31 km from the coast of the Caspian Sea		3.5 mln pcs	

Note. *— Since 2009, all other hatcheries—branches of the North Caspian Basin Administration for Fisheries and Conservation of Aquatic Biological Resources “Sevkasprybvod”); **—now: Scientific and Experimental Complex of Aquaculture—SECA “BIOS”, Volga-Caspian Branch of the Federal State Budgetary Scientific Institution “All-Russian Research Institute of Fisheries and Oceanography (“CaspNIRH”)”.

**Table 2 biology-11-01240-t002:** Types of morphological abnormalities in the structure of body and external organs in sturgeons, (+)—presence, n/d—no data.

Abnormalities Type	Species						
	Siberian Sturgeon	Russian Sturgeon	Persicus Sturgeon	Beluga Sturgeon	Stellate Sturgeon	Sterlet	Fringebarbel Sturgeon
Body shape	+	+	n/d	+	+	+	+
Fins	+	+	n/d	+	+	+	+
Opercula	+	+	n/d	+	n/d	n/d	+
Olfactory organs	+	+	n/d	+	n/d	n/d	+
Eyes	+	+	n/d	+	n/d	n/d	+
Mouth	+	+	n/d	+	n/d	n/d	n/d
Gametogenesis	+	+	n/d	+	n/d	n/d	+
Muscle	n/d	_+_	n/d	n/d	+	n/d	+

## Data Availability

Not applicable.

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
