# Peer review of "Experience, Principles and Parameters in the Sturgeon Quality Assessment by Anomalies in Early Ontogenesis (A Review)"

_biology, 2022, doi:10.3390/biology11081240_

Round 1
Reviewer 1 Report
The 2nd manuscript is of much better quality than the first. It has improved both in its English language and in its content.
It would further improve the quality of the manuscript if the authors would edit a new figure (either with photographs or drawings) to illustrate the anomalies observed. It would help readers to identify the observed anomalies! This is sorely lacking in the manuscript! For example instead of Table 2, because in my opinion it does not contain really important information. Half of the table just says "no data". So in my opinion it is not needed.
A detailed list of errors:
In the title I would definitely indicate that these are observations based on Kazakhstan's experience. So "Experience, Principles and Parameters in the Sturgeon Quality Assessment by Anomalies in Early Ontogenesis in Kazakhstan (a Review)"
or: " in Caspian countries"!
Line 29-30: This sentence should not be read as "The most 29 frequently recorded morphological anomalies." Something is missing.
Line 67: I think "were found" is not the correct term here.
Line 89-90: In the PDF version, the Russian text remains, and in the References, in several places.
Line 114: The abbreviation is not "ICUN" but IUCN!
In column 5 of Table 1, either type "mln pcs" next to each figure or not, but make it uniform!
In Reference 4th, the font and size are not correct.
Author Response
Dear reviewer!
Answer in attachment.

Reviewer 2 Report
The changes made to the text improve the understanding of the presented study. I did not indicate anything for improvement in my review, so I confirm what I said.
Author Response
Dear reviewer!
Answer in attachment.

This manuscript is a resubmission of an earlier submission. The following is a list of the peer review reports and author responses from that submission.
Round 1
Reviewer 1 Report
It is not at all clear to the reviewer exactly what the subject of this “review” article would be. The article contains well-known generalizations about the declining occurrence of sturgeon in natural waters, and that this can be attributed to human activity. It is also evident that the hatchery propagation is needed partly due to market demands and partly due to reintroduction into natural waters. The only new information in the article is the number of sturgeon breeding companies in the Caspian Basin. But in addition, we do not know any other information. The manuscript contains many assumptions and factual errors, such as Line 151. “imported fertilized eggs in most cases are of hybrid origin”. Yes, the companies produce hybrids for caviar production, but you can also buy pure bred stock of the same species. The other problematic part is the issue of larval deformity. The percentage of larval deformity in a well-functioning fish farm is 1-2% for sturgeon larvae. Higher rate (100% ????) qualifies the breeding farm. It refers to serious professional misconduct. Artificial reproduction of sturgeon is not a biotechnology and has already been developed. It is used in many fish farms.
So the manuscript does not contain any information that could be of interest.
About the quality of the manuscript: The English language of the manuscript needs to be developed very strongly. The fine arts part in the introduction is completely redundant. The one part of 2. “Materials and Methods” rather is “Introduction” and the other part is “Results”.
The “%”sign is in the title of the "Table 2.", but there is no percentage anywhere in the table, only “+” signs and “n/a”. If it contained %, it would make sense to include this Table in the manuscript.
Reviewer 2 Report
The Authors of the review titled "Experience, Principles and Parameters the Sturgeon Quality Assessing by anomalies in early ontogenesis (a Review)" have prepared an extremely underwhelming paper, which appears to be a raw draft rather than a finished manuscript.
Most importantly, the quality of the English language throughout the text is extremely poor, which is why I decided to finish my specific line-by-line review already after the first section ("Introduction"). Everything is lacking in this regard - grammar, vocabulary, overall style. In some cases, the occurring errors cause sentences to be illogical or even totally wrong (sometimes they are simply absurd or even funny). Someone with a much better English language proficiency has to review this paper, sentence by sentence.
Furthermore, already at the beginning the Authors start to jump randomly from archaeology, to art history, then all of a sudden to modern times and sturgeon aquaculture. In my opinion, the composition (order) of these topics feels very amateurish, unfortunately. I believe that focusing on modern times and the current situation of wild and farmed sturgeons should be a desirable approach, rather than these unnecessary jumps from idea to idea. This issue remains afterwards prevalent throughout the whole manuscript - it is very hard to actually grasp what the purpose of this review actually was. Is it only a locally-oriented review (Russia and Kazakhstan), or not?
Furthermore, the Authors did not introduce proper section names, but sticked to the default "Materials and Methods", what is of course another mistake. Besides, being tagged as a "review", this paper is relatively short and contains only 54 references - is that truly everything that could be found within this field of study?
All in all, this paper needs some serious improvements before a more in-depth analysis of its contents can be performed. I have aligned my specific commentary (reaching the end of the Introduction) below, and have also attached the same comments in the edited .pdf file.
Title: There is something missing here, likely "in" before "the". Also, change to noun "Assessment".
Abstract: To be brutally honest, this section is quite terrible. Below, I have suggested some of the most-necessary changes which need to be introduced, but I believe that the Authors should simply rework this section from scratch, starting with the removal of excessive passive voice ("... was made.", "... is stated.", "... is given." etc.). There is also no indication of the context of this review, just plain statements about some parts of the content, all written in a very chaotical manner. In general, the style of every sentence has to be improved (totally rewritten). As a potential Reader I would be instantly repelled from this article already after a few sentences of this abstract.
Lines 14-15: Please delete this whole sentence.
Lines 16-19: How does all of this "belong" to Russian scientists?
Line 19: Delete "There is practice of".
Line 21: Delete "Gegenetic".
Line 25: Delete "resulting".
Line 27: "Corps"? What is that? You mean whole body deformities? Furthermore, "violations" is definitely misused here.
Line 28: "Properties" of what?
Line 29: "Clade" is not a term which should be used here, rather "classes".
Introduction: I have only highlighted some major spotted mistakes, but literally every sentence needs to be reworked in this section, unfortunately.
Lines 39-40: "era as it is thought to be about 250 million years ago" - improve the style and positioning.
Line 42: "Evolutions"?
Line 44: Delete "of the geological history of the Earth".
Line 45: "Over next 186 million years, the sturgeon aquaculture has reached its zenith" - what on Earth is that supposed to mean? Sturgeon aquaculture exists since 186 million years? Who cultured them, dinosaurs?
Line 46: Aquaculture "retains features"? The linguistic mistakes just keep mounting on, resulting in more and more ridiculously sounding statements.
Line 47: Chord? Furthermore, sturgeons do not lack vertebrae, but vertebral centra. This is yet another huge mistake.
Lines 49-50: Delete this whole part "for more than 160 million years, these cartilaginous ganoids have been living as "living fossils."".
Lines 52-69 and Figure 1: This part serves nothing, I suggest to delete it entirely.
Line 71: "Fishing epithets" - what?
Line 80: Delete "been".
Line 85: Delete "an".
Line 86: "including in Kazakhstan" - including what?
Lines 90-95: The context of this sentence is somewhat understandable, but the style is very poor.

Reviewer 3 Report
The manuscript is well organized and the description of the review is clear and the text is well readable. The scenario has been observed through an extensive bibliography and offers a good overview. The discussion can be useful to raise awareness in product quality management.